# hTERT Downregulation Attenuates Resistance to DOX, Impairs FAK-Mediated Adhesion, and Leads to Autophagy Induction in Breast Cancer Cells

**DOI:** 10.3390/cells10040867

**Published:** 2021-04-10

**Authors:** Aleksandra Romaniuk-Drapała, Ewa Totoń, Natalia Konieczna, Marta Machnik, Wojciech Barczak, Dagmar Kowal, Przemysław Kopczyński, Mariusz Kaczmarek, Błażej Rubiś

**Affiliations:** 1Department of Clinical Chemistry and Molecular Diagnostics, Poznan University of Medical Sciences, 49 Przybyszewskiego St., 60-355 Poznań, Poland; aromaniuk@ump.edu.pl (A.R.-D.); etoton@ump.edu.pl (E.T.); koniecznanatalia2@gmail.com (N.K.); dagmar.kowal@abbott.com (D.K.); 2Department of Cancer Immunology, Poznan University of Medical Sciences, 60-806 Poznan, Poland; marta1machnik@gmail.com; 3Department of Head and Neck Surgery, Poznan University of Medical Sciences, The Greater Poland Cancer Centre, 61-866 Poznan, Poland; barczak.woj@gmail.com; 4Centre for Orthodontic Mini-Implants at the Department and Clinic of Maxillofacial Orthopedics and Orthodontics, Poznan University of Medical Sciences, 60-812 Poznan, Poland; pkopczynski@orto1.net; 5Department of Immunology, Chair of Clinical Immunology, Poznań University of Medical Sciences, 5D Rokietnicka St., 60-806 Poznań, Poland; markacz@ump.edu.pl

**Keywords:** hTERT, RNAi, adhesion, migration, senescence, autophagy, breast cancer

## Abstract

Telomerase is known to contribute to telomere maintenance and to provide cancer cell immortality. However, numerous reports are showing that the function of the enzyme goes far beyond chromosome ends. The study aimed to explore how telomerase downregulation in MCF7 and MDA-MB-231 breast cancer cells affects their ability to survive. Consequently, sensitivity to drug resistance, proliferation, and adhesion were assessed. The lentiviral-mediated human telomerase reverse transcriptase (hTERT) downregulation efficiency was performed at gene expression and protein level using qPCR and Western blot, respectively. Telomerase activity was evaluated using the Telomeric Repeat Amplification Protocol (TRAP) assay. The study revealed that hTERT downregulation led to an increased sensitivity of breast cancer cells to doxorubicin which was demonstrated in MTT and clonogenic assays. During a long-term doubling time assessment, a decreased population doubling level was observed. Interestingly, it did not dramatically affect cell cycle distribution. hTERT downregulation was accompanied by an alteration in β1-integrin- and by focal adhesion kinase (FAK)-driven pathways together with the reduction of target proteins phosphorylation, i.e., paxillin and c-Src. Additionally, autophagy activation was observed in MDA-MB-231 cells manifested by alternations in Atg5, Beclin 1, LC3II/I ratio, and p62. These results provide new evidence supporting the possible therapeutic potential of telomerase downregulation leading to induction of autophagy and cancer cells elimination.

## 1. Introduction

One of the most important directions in the development of targeted cancer therapy is to obtain specificity of action with limited side effects. Among cancer cells’ critical features, we can list the migratory and invasion potential, resistance to therapy, immortality, or ability to escape from cell death. It is supposed that most of these processes can be driven by or at least associated with telomerase [1]. The enzyme’s main function is the maintenance of telomere length that prevents DNA degradation, chromosomal fusion (potentially leading to polyploidization and cell death), and DNA repair activities [2]. However, in normal cells, these structures degrade with each cell cycle/division due to the transcriptional repression of the key enzyme’s subunit, human telomerase reverse transcriptase (hTERT), which takes place during early embryonic development [3]. Consequently, telomere shortening induces replicative senescence or may eventually lead to cell death [4,5,6,7]. However, telomerase can be restored in cancer cells and provide unlimited replicative potential. In more than 90% of cancer cases, hTERT is overexpressed, and its activation is a fundamental step in tumorigenesis [8]. Additionally, there is increasing evidence for the telomere-unrelated roles of hTERT in tumor cells. Presumably, some noncanonical functions are associated with modifications of drug resistance, proliferation, and adhesion abilities of cancer cells that may affect their survival [9,10,11,12].

The cell adhesion to the extracellular matrix (ECM) plays a critical role in regulating essential cellular functions, including the mechanisms mediated by focal adhesion kinase (FAK) [13,14]. FAK is a cytoplasmic non-receptor tyrosine kinase that is overexpressed in many cancers, including glioblastoma, breast cancer, colorectal cancer, pancreatic cancer, lung cancer, ovarian cancer [15]. Phosphorylation of FAK at tyrosine 397 plays an essential role in tumor cell signaling and can be induced by growth factors and mechanical stress. In turn, autophosphorylation of FAK at Tyr397 (Y397) generates a binding site for c-Src, which phosphorylates FAK at Tyr576 (Y576 FAK) and Tyr577 (Y577 FAK). These changes promote maximal FAK catalytic activity associated with the metastatic status of tumors [16].

Autophagy is a highly conserved cellular process by which defective organelles, non-functional proteins, and lipids become sequestered within structures called autophagosomes. They fuse with lysosomes, and the engulfed components are then degraded by lysosomal enzymes [17]. The mechanism of this process is very complex. It is regulated by several factors, such as the mTOR signaling pathway, P70S6 kinase, PI3 kinase type I and III, beclin-1, Atg5, or the protein DAPK [18]. Recent observations suggest that autophagy is essential in regulating survival and death signaling pathways in various human diseases, including cancer. Numerous studies indicate that autophagy induction after anticancer therapy may lead to survival or death of cancer cells. Some drugs may induce this process in breast cancer cells, but it is difficult to predict which factors tip the scales [19,20].

As demonstrated by Noureini et al., administration of chelidonine in MCF7 cells at very low concentrations induced apoptosis and at high autophagy, and this effect was accompanied by suppressed hTERT expression [21]. Further experiments revealed a significant role of hTERT translocation to mitochondria in response to stress, autophagy activation, and mitochondria metabolism. Interestingly, it was shown that autophagy activation was also associated with decreased levels of mitochondrial DNA damage [22]. Another study showed more detailed results on the role of hTERT in activation of autophagy and improved mitochondrial function in aged hepatocytes [23]. Altogether, it seems that the contribution of hTERT/telomerase to cell survival depends on the metabolic conditions, including hypoxia, aging, and stress. Nevertheless, this aspect remains elusive, also in breast cancer cells.

Recent observations reveal multiple oncogenic activities of telomeric subunits hTR and hTERT, including contribution to DNA damage repair, mitochondrial function, gene transcription, metastasis, as well as autophagy [9,24,25,26,27]. These reports indicate how important the enzyme is in a broad range of molecular processes [28]. To date, several strategies for using telomerase inhibition to eliminate a wide variety of human malignancies have been proposed. 

This report contains efficient and stable hTERT downregulation with lentiviral particles in a model of two breast cancer cell lines MCF7 and MDA-MB-231. The study aimed to investigate how telomerase regulation affects breast cancer cell metabolism in the context of metastatic potential that is mainly driven by adhesion and migration capability. For this reason, a stable transduction system was chosen for a long-term hTERT downregulation, especially since the correlation between hTERT and cancer cell motility and tumorigenicity remains unclear [29,30,31]. 

## 2. Materials and Methods

### 2.1. Cell Culture

Two cell lines representing different molecular subtypes of breast cancer were enrolled in the study, i.e., MCF7 (ER/PR+, HER2-, Ki-67-, TP53WT) and MDA MB-231—basal-like subtype, also called triple-negative breast cancer (TNBC; ER/PR-, HER2, TP53mut). MCF-7 cell line, in comparison to the MDA-MB-231 cell line, is a poorly aggressive and non-invasive cell line. Overall, it is being considered to have low metastatic potential. In contrast, MDA-MB-231 is a highly aggressive, invasive, and poorly differentiated cell line. The MCF-7 cell line has proven to be a useful model of hormone-responsive breast cancer. It is a particularly valuable model in preclinical testing of antiestrogen therapies (e.g., tamoxifen and aromatase inhibitors) and identifying resistance mechanisms to such drugs. In turn, the MDA-MB-231 cell line is not sensitive to trastuzumab, HER2 targeted treatment, and tamoxifen, which is part of endocrine therapy.

The cells were maintained in RPMI-1640 (Biowest, Nuaillé, France) medium supplemented with 10% fetal bovine serum (FBS) (Biowest, Nuaillé, France) at 37 °C in an atmosphere of 5% CO2 and saturated humidity. Both human breast adenocarcinoma cell lines, MCF7 (HTB-22) and MDA-MB-231 (HTB-26), were obtained from the American Type Culture Collection (ATCC).

### 2.2. Lentiviral Vector Production, Titration, and Transduction

The 2nd generation system was used to produce lentiviral vectors (pLV-THEM-shTERT and control vector pLV-THEM-shRNA). The HEK-293T cells were co-transfected with packaging plasmids psPAX2 (#12260, Addgene, Watertown, MA, USA), VSV-G-expressing envelope plasmid pMD2.G (#12259, Addgene, Watertown, MA, USA), and lentiviral plasmid pLV-THEM-GP1 (#12247; Addgene, Watertown, MA, USA). The production, transduction, and titration were carried out according to the protocols described in Szulc et al., 2008 and Barczak et al., 2014 [32,33]. Briefly, the culture supernatant was collected 48 h post-transfection and passed through 0.45-μm filters, concentrated, and aliquots were stored at −80 °C. All breast cancer cell lines (2 × 10^4^) were seeded in a six-well plate 24 h before transduction. Cells were transduced with lentiviral shRNA targeting hTERT with polybrene (5 μg/mL; Sigma-Aldrich, St. Louis, MO, USA) in a growth medium with reduced FBS concentration (5%). Cells infected with non-silencing lentiviral shRNA served as controls. The media was replenished after 48 h, and 3 μg/mL puromycin (Sigma-Aldrich, St. Louis, MO, USA) was added 5 days after infection. The cells were selected using puromycin for 5 days and were subsequently tested for *hTERT* expression. The green fluorescent protein (GFP) transgene expression was observed by an inversion fluorescence microscope (Axiovert, Carl Zeiss, Jena, Germany). The excitation wavelength of the blue laser was 488 nm, and the detection wavelength was 520 nm. Noteworthy, all the experiments were performed on day 21 from transduction. That was the first time point to show significant alterations in hTERT expression at all tested levels, i.e., mRNA, protein, and the whole telomerase complex activity. Additionally, we were interested in the assessment of a long-term effect of hTERT downregulation. In turn, longer culture led to radical inhibition of cell growth that made further metabolic tests more difficult and less reliable due to some non-specific effects.

### 2.3. Real-Time PCR Analysis

Quantitative analysis of hTERT gene expression was assessed using qPCR. The MCF7 and MDA-MB-231 cells treated with pLV-THEM-shTERT (shRNA hTERT) and pLV-THEM-shRNA (shRNA control) (5 × 10^5^) were seeded into 60 mm plates. After 48 h, total RNA isolation was done using the High Pure RNA Isolation Kit (Roche, Basel, Switzerland) according to the manufacturer’s protocol. Concentration and quality ratios (A260/A280) of extracted RNA were evaluated by optical density measurement with Biophotometer Plus (Eppendorf, Hamburg, Germany). cDNA was synthesized with Transcriptor First Strand cDNA Synthesis Kit (Roche, Basel, Switzerland) using 1 µg of total RNA, oligo dT primers, and random hexamer primers. The real-time PCR was carried out using LightCycler 96 (Roche, Basel, Switzerland) with specific primers obtained from the commercial set (Real-Time Primers, St. Louis, MO, USA). Amplification products of individual gene transcripts were detected with LightCycler^®^ FastStart Essentials DNA Green Master (Roche, Basel, Switzerland). The reaction conditions were as follows: 95 °C for 10 min; (94 °C for 15 s; 60 °C for 15 s; 72 °C, 15 s) × 40; 72 °C for 5 min. The GAPDH expression was provided as an internal reference gene (housekeeping gene) to normalize the expression of the hTERT.

#### Relative Telomere Length Assessment Using qPCR

DNA was extracted from cancer cells (1 × 10^6^ cells in each sample) after hTERT downregulation using a Genomic Mini DNA Isolation kit (A&A Biotechnology, Poland). A high concentration sample of genomic DNA was prepared in decimal concentrations that were used to run as a standard curve. Telomere length was assessed using two pairs of primers, specific towards telomeres (Telg: ACACTAAGGTTTGGGTTTGGGTTTGGGTTTGGGTTAGTGT and Telc: TGTTAGGTATCCCTATCCCTATCCCTATCCCTATCCCTAACA) [34] and single copy gene, albumin (ALBF: TTTGCAGATGTCAGTGAAAGAGA and ALBR: TGGGGAGGCTATAGAAAATAAGG), as previously described [35]. Briefly, the conditions were optimized as follows: 95 °C for 10 min, followed by two cycles of 94 °C for 15 s and 49 °C for 15 s without fluorescence acquisition and 40 cycles (94 °C for 10 s, 61 °C for 10 s and 72 °C for 10 s) with signal acquisition. For albumin gene copies the conditions were optimized as follows: denaturation at 95 °C for 10 min, followed by 45 cycles at 94 °C for 10 s, 61 °C for 10 s, and 72 °C for 10 s. The MgCl_2_ was 2.5 mM in both reactions while the primer concentration was 0.9 or 0.5 µM for telomere or albumin copies assessment, respectively. Melting analysis (range, 65–95 °C; resolution, 0.2 °C) was performed to verify the specificity of the products. The efficiency of the reactions was higher than 95%. The assay was performed using the LightCycler^®^ 2.0 Instrument and the LightCycler^®^ FastStart DNA Master SYBR Green I kit (Roche, Basel, Switzerland).

### 2.4. Western Blot Analysis

Cells were treated with pLV-THEM-shTERT (shRNA hTERT) and pLV-THEM-shRNA (shRNA control) and doxorubicin (Sigma-Aldrich, St. Louis, MO, USA) as indicated. The MCF7 and MDA-MB-231 cells were seeded at a density of 1 × 10^6^ cells into 100 mm culture plates, incubated overnight, and allowed to attach. Then, On the 21st-day, post-transduction doxorubicin (DOX) was added at a concentration 0.1 μM for 8 h. Total protein lysates were extracted with RIPA buffer (Thermo Scientific, Waltham, MA, USA). The protein concentration in the sample was measured using Bradford assay (Sigma-Aldrich, St. Louis, MO, USA), according to the manufacturer’s protocol, and 40 µg of total protein of each cell extract was loaded onto SDS–PAGE gels. Western blot was performed by a standard procedure using a Polyvinylidene fluoride (PVDF) membrane (Thermo Scientific, Waltham, MA, USA). Non-specific binding was blocked by incubation in 5% non-fat milk in Tris-buffered saline and Tween 20 at room temperature for 1 h. The following primary antibodies were used for detection: anti-hTERT (1:1000, Novus Biologicals, Centennial, CO, USA), anti-p21, anti-β1-integrin, anti-FAK, anti-p-FAK Y397, anti-p-FAK Y576/577, anti-paxillin, anti-p-paxillin, anti-Src, anti-p-Src Y527, anti-p-Src Y416, anti-Atg5, anti-p62, anti-LC3 I/II, anti-mTOR (1:1000, Cell Signaling Technology, Danvers, MA, USA); anti-Ki-67, anti-p53, anti-p-Ser15 p53, anti-beclin-1 (1:1000) and anti-GAPDH (1:3000, Santa Cruz Biotechnology, Santa Cruz, CA ,USA). After removing the antibodies, anti-rabbit IgG or anti-mouse IgG (1:1000, Cell Signaling Technology, Danvers, MA, USA) secondary antibodies labeled with horseradish peroxidase were added. The proteins were detected by Super Signal West Pico Chemiluminescent Substrate (Thermo Scientific, Waltham, MA, USA) or ECL™ Prime Western Blotting System (GE Healthcare, Waukesha, WI, USA) using camera and VisionWorks software (UVP, Inc., Upland, CA, USA). Additionally, results were analyzed semi-quantitative using Image Studio Lite (LI-COR Biosciences, Lincoln, NE, USA). 

### 2.5. Telomerase Activity-TRAP

The effect of hTERT downregulation on telomerase activity was assessed using the quantitative TRAPEZE^®^ RT Telomerase Detection Kit (Merck Millipore, Darmstadt , Germany), according to the manufacturer’s instructions. This assay is based on the telomeric repeat amplification protocol (TRAP) as previously described [21]. Briefly, on day 21, post-transduction of MCF7 and MDA-MB-231 cells with pLV-THEM-shTERT (shRNA hTERT) or pLV-THEM-shRNA (shRNA control), total cellular proteins were extracted using CHAPS lysis buffer. For each assay, 1 µg of total protein extract was used. The protocol consisted of a telomerase-primer elongation reaction, followed by 45 PCR cycles. The results were quantitated using fluorescein-labeled Amplifluor^®^RP primers. A standard curve was prepared as a dilution series of TSR8 control templates. Heat-inactivated cell extracts and lysis buffer were used as a negative control. The amount of elongated telomerase substrate produced in each well from the telomerase activity was determined from a linear function of log10 of the attomoles of TSR8 control standards (number of repeats) versus the Ct values. The mean value of the three replicates from separate wells for each sample was calculated.

### 2.6. MTT Cell Survival Assay

Cell survival was determined using MTT assay by assessing the sensitivity of cells subjected to hTERT downregulation to anticancer drug doxorubicin (DOX). The MCF7 and MDA-MB-231 cells were seeded at a density of 5 × 10^3^ cells per well in 96-well culture plates, incubated overnight to allow for cell attachment, and then on 21st-day post-transduction DOX was added at a concentration range of 0–5 μM. Cells were treated at the required times (24–72 h) and incubated with 10 μL MTT reagent (5 mg/mL) (Sigma-Aldrich, St. Louis, MO, USA). The cells were incubated at 37 °C for 4 h, followed by 100 μL of solubilization buffer (10% SDS in 0.01 M HCl) addition. The absorbance was measured in each well with the Microplate Reader Multiskan FC (Thermo Scientific, Waltham, MA, USA) at two wavelengths of 570 and 690 nm. Each experimental point was determined in triplicate. IC_50_ (half-maximal (50%) inhibitory concentration) values were calculated using CompuSyn (ComboSyn, Inc. Paramus, NJ, USA), and the standard deviation was calculated using Excel software (Microsoft, Syracuse, NY, USA).

### 2.7. Colony Formation Assay

The clonogenic assay was used to confirm the effectiveness of hTERT downregulation on the sensitization of cells to DOX, as previously described [36]. Briefly, the MCF7 and MDA-MB-231 cells were seeded into 60 mm plates in the concentration of 200 cells per well. After overnight incubation in standard conditions, on 21st-day post-transduction, cells were exposed to 10–500 nM of DOX. After 24 h, the medium was removed, and cells were washed twice with phosphate-buffered saline (PBS). Then the fresh medium was added, and cells were maintained for 14 days, with media change every four days. After that time, a 10 min fixation in methanol and staining with 1:20 aliquot of Giemsa’s stain (Sigma-Aldrich, St. Louis, MO, USA) for 1 h were performed. The wells were washed with distilled water, air-dried, and the colonies were enumerated. The experiment was repeated three times for each cell line [36].

### 2.8. Cell Cycle Analysis

In order to analyze the influence of long-term hTERT downregulation on the cell cycle, a flow cytometry analysis using propidium iodide was performed, as previously described [36]. Briefly, the MCF7 and MDA-MB-231 cells were seeded into 6-well plates in the concentration of 1 × 10^5^ cells per well. After 48 h incubation in standard conditions, on 21st-day post-transduction, cells were harvested, washed twice with phosphate-buffered saline (PBS), then incubated with a solution of 0.05% saponin, 50 µg/mL propidium iodide, and 10 mg/mL of RNase A (Sigma-Aldrich, St. Louis, MO, USA) in PBS for 1 h at 37 °C. DNA content was analyzed by flow cytometry at the emission wavelength of 488 nm using FACScan (Becton–Dickinson, Franklin Lakes, NJ, USA). The relative proportions of cells in the G1/G0, S, and G2/M phases of the cell cycle were determined from the obtained data. Three separate experiments in triplicates were performed for each cell line.

### 2.9. Assessment of Population Growth

The MCF7 and MDA-MB-231 cells treated with pLV-THEM-shTERT (shRNA hTERT) and pLV-THEM-shRNA (shRNA Control) during 9 weeks were included in the assessment of population growth. Every Monday, cells were seeded into 100 mm plates in the concentration of 3.5 × 10^5^ for MCF7 and 4.5 × 10^5^ for MDA-MB-231 cells per well. After 96 h, cells were passaged and counted using the Fuchs-Rosenthal chamber and Axiovert 40 CFL microscope (Carl Zeiss, Jena, Germany). For every 96 h, cumulative population doublings (CPD) and doubling time t_2/1_ were calculated. Population doubling level is the total number of times the cells in a given population have doubled during in vitro culture.
CPD = (log (total number of cells counted at the day of passage) − log (number of cells initially seeded at the previous passage))/log2 
t_2/1_ = ln_2/r_ (h)

### 2.10. Senescence-Associated SA-β-Galactosidase Assay

In order to explain changes observed in the cell cycle and to analyze the influence of long-term hTERT downregulation on the induction of cellular senescence, SA-β-galactosidase staining (Sigma-Aldrich, St. Louis, MO, USA) was performed. At week 7 post-transduction, the MCF7 and MDA-MB-231 cells were seeded into 6-well plates in the concentration of 5 × 10^4^ cells per well. After 72 h incubation in standard conditions, cells were washed with PBS, fixed in 4% paraformaldehyde for 20 min at room temperature. The cells were rewashed with PBS twice and immersed for 4 h in the staining solution at 37 °C. The senescent cells were identified as green-stained cells under inverted microscope Axiovert 40 CFL (Carl Zeiss, Jena, Germany) according to the manufacturer’s instruction (magnification 100×).

### 2.11. Cell Adhesion Assessment

On the 21st-day post-transduction, the MCF7 and MDA-MB-231 cells (5 × 10^5^) were seeded into 60 mm plates coated with Matrigel^®^ (Corning, New York, NY, USA). Cells were analyzed under the phase-contrast microscope (Carl Zeiss, Jena, Germany) and photographed after 15, 30, 60 min, 3 h. After 3 h incubation, the medium with detached cells was replaced. The pictures are representative of three independent experiments (magnification 40×). Adhesion cells were counted from at least three fields in each well. Each bar represents the mean ± SD of the data obtained from three independent experiments. 

### 2.12. Wound-Healing Migration Assay

For studying cell migration abilities after hTERT downregulation, a scratch wound healing assay was performed. The MCF7 and MDA-MB-231 cells were seeded into 6-well plates in the concentration of 1 × 10^5^ cells per well. On 21st-day post-transduction, on the monolayer, a cross-shape scrape was made with a P-200 pipette tip; then, the medium was replaced. The wounded areas were marked for observation and photographed at the indicated time (24, 48, and 72 h) after scratch (magnification 100×). The micrographs of the scratch wound healing assay are representative of three independent experiments. The migrated cells were quantified by measuring wound closure areas after injury—all cells were counted in this area. Each bar represents mean ± SD (n = 3). The results were shown relative to the shRNA Control sample.

### 2.13. Statistical Analysis

Results were expressed as mean ± SD. All statistical analyses were carried out using GraphPad Prism (GraphPad Software, Sandiego, CA, USA). Differences were assessed for statistical significance using repeated-measures ANOVA, followed by post-hoc the Dunnett’s test method. All experiments were performed in triplicates unless specified otherwise. The threshold for significance was defined as p < 0.05 and are indicated by the (*) symbol for *p* < 0.05, by (**) for *p* < 0.01, by (***) for *p* < 0.001.

## 3. Results

### 3.1. hTERT Downregulation

Telomerase (or hTERT) downregulation is associated with cell survival inhibition. Alternatively, some studies show that hTERT (in a PI3K/AKT/mTOR pathway-dependent manner) reveals a pro-survival effect [37,38]. For this reason, the first stage of our study included generating a stable and efficient transduction protocol to observe long-time effects of hTERT downregulation. We used an in vitro model of breast cancer cells. The selection of cell lines was supported by their molecular characteristics, as described in the Materials and Methods section. According to our studies, selected cell lines, i.e., MCF7 and MDA-MB-231, show moderate but similar hTERT expression levels. Another crucial difference is that the MCF-7 cell line is CASP-3-deficient. Additionally, in comparison to the MDA-MB-231, it is a significantly less aggressive and less invasive type of cells. From the phenotypic point of view, it means such cells would show significantly lower metastatic potential. Since the contribution of hTERT to the migration and invasion potential of cancer cells is postulated [26,30], both cell lines were enrolled in the study. 

The cells were transduced with lentiviral particles. The transduction efficiency (the mean fluorescence intensity reflecting GFP transgene expression) were determined by flow cytometry and verified using a microscope (Axiostar, Carl Zeiss, Jena, Germany; Figure 1A) after puromycin selection. Both experiments revealed a very high rate of transduction efficacy. 

hTERT downregulation and cell growth were being monitored for up to 45 days. However, all the experiments were performed on day 21 from transduction, as mentioned in the Materials and Methods section. Since some reports postulate the additive effect of therapeutic agents and telomerase modulators in cancer cells, we included tests using a combination of hTERT downregulation and doxorubicin. The concentration of the drug was carefully selected based on previous MTT experiments [36], and it was 0.1 μM.

A significant reduction of hTERT protein level at 75% (*p* < 0.01) in MCF7 and 60% (*p* < 0.05) in MDA-MB-231 cells was observed after hTERT gene downregulation (Figure 1B). Importantly, doxorubicin alone in control cells or shRNA hTERT cells did not cause any significant effect in hTERT expression (western blot) when subjected to the treatment in both tested cell lines (Figure 1B). Similarly, no additive effect in the level of protein accumulation was observed. Further, hTERT downregulation was shown effective in reducing the hTERT transcript level in both MCF7 and MDA-MB-231 cells (reduction by 60%, *p* < 0.01 and 70%, *p* < 0.01, relative to mock shRNA, respectively) (Figure 1C). Similarly, telomerase assessment by TRAP assay revealed a significant decrease of the enzyme activity in both cell lines >75% after hTERT downregulation (*p* < 0.01, Figure 1D).

We expected that telomerase downregulation would eventually lead to telomeres attrition so we performed the assessment of their length dynamics. After 21 days from transduction, telomere length in MCF7 cells was significantly reduced by more than 25% relative to shRNA Control sample. In MDA-MB-231 cells, the reduction was not significant (lower than 10%), but also noticeable (Figure 1E).

### 3.2. The Effect of hTERT-Downregulation on Breast Cancer Cells Sensitivity to Doxorubicin

It is hypothesized that an approach based on telomerase downregulation or inhibition combined with adjuvant therapeutic agents, such as chemotherapeutics may enhance tumor suppression. Thus, both transduced cell lines (21 days after transduction) were treated with increasing concentrations of doxorubicin, and the cell viability was assessed by MTT. 

It was found that hTERT downregulation efficiently increased cancer cell lines’ sensitivity to doxorubicin in a time- and dose-dependent manner. As shown in Figure 2, hTERT-downregulated cells appeared to be more sensitive to the drug compared to shRNA Control cells. In the case of a 24-h treatment, a statistically significant reduction in the survival of hTERT shRNA-treated cells for both cell lines was observed only at the highest concentration of doxorubicin used, 5 μM (circa 60 vs. 40% reduction in MCF7, and 55 vs. 40% in MDA-MB-231; *p* < 0.05). After a 48-h treatment, the significant differences were revealed already at the concentration of 0.5 μM (and higher) with the observed reduction of the survival in MCF7 (55 vs. 40% reduction) and MDA-MB-231 (50 vs. 30% reduction) relative to control cells (*p* < 0.01 and *p* < 0.05, respectively). For 72 h of incubation, a significant reduction in survival of the MCF7 hTERT shRNA cells in relation to control cells was observed already after treatment with 0.05 µM (75 vs. 60%; *p* < 0.05). Similar changes in the MDA-MB-231 shRNA hTERT cells were noticed only at 1 μM DOX (70 vs. 50%; *p* < 0.01) and higher (Figure 2A,B). 

### 3.3. The Effect of Doxorubicin and hTERT Downregulation on Breast Cancer Cells Colony Formation

To analyze the effect of hTERT silencing on the sensitization of MCF7 and MDA-MB-231 cells to doxorubicin in the genotoxic context, a clonogenic assay was performed. The DOX concentration range was chosen based on the MTT assay results (0, 10, 50, 100, or 500 nM). The clonogenic assay is the method of choice to verify cell survival. The reduction of telomerase expression decreased the colony formation capacity in both examined breast cancer cell lines (Figure 3A,B). In MCF7, the difference was significant in the whole range of applied concentrations (*p* < 0.05). In contrast, in MDA-MB-231 cells, the difference between control and hTERT-downregulated cells was noticeable only at the DOX concentration 50 nM and higher. The observed significant increase in the sensitivity of hTERT-downregulated MCF7 to DOX is evident, confirming MTT results.

### 3.4. The Contribution of hTERT Silencing to Cell Cycle Modification in MCF7 and MDA-MB-231 Cells

Since hTERT downregulation eventually leads to cancer cell death, we wanted to verify if hTERT downregulation affected the breast cancer cell cycle distribution 21 days after transduction. For this reason, we performed a flow cytometry analysis of the cell cycle using propidium iodide labeling (histograms presented in Figure 4A). In the case of MCF7 shRNA hTERT cells, significant accumulation of cells in the G0/G1 phase (increase by about 10%; *p* < 0.05) was observed relative to control cells (shRNA Control) (Figure 4B). Simultaneously, there was a reduction in the number of cells in the G2/M phase to 10% (*p* < 0.05), compared to 16% in control cells. However, no significant changes in the number of other subpopulations were observed. Interestingly, in the MDA-MB-231 hTERT shRNA samples, we observed a two-fold increase in the percentage of dead cells (from 3% to 6%; *p* < 0.05), but the level was still low (Figure 4B).

Additionally, we assessed caspase-3 activation in target cells but only in MDA-MB-231 since MCF7 does not show functional caspase-3 due to genomic mutation [39]. We used 2 uM DOX (8 h treatment) as a positive control for apoptosis assessment (Figure 4C). Importantly, hTERT downregulation did not provoke any significant procaspase-3 cleavage. Subcytotoxic concentration of DOX (0.1 μM) provoked slight but significant apoptosis induction, but it did not provoke any significant cumulative effect in shRNA hTERT cells.

### 3.5. The Influence of hTERT Knockdown on the Proliferation Potential of Breast Cancer Cells

Since we could not notice any dramatic alteration in the cell cycle of both cell lines after 21 days from transduction and no significant induction of caspase-3 (in MDA-MB-231 cells), we performed experiments to verify any modulation of the proliferative potential of studied breast cancer cells. During the long-term experiment, both cell lines were monitored for proliferative potential after the hTERT was downregulated. Starting from the transduction day, for each 96 h growth interval, doubling time (t_2/1_) and growth factor (r) were calculated. The MCF7 hTERT shRNA cells during the 9-week evaluation period revealed a decreasing population doubling level. Due to the hTERT silencing, the doubling time increased gradually from 36 h to 108 h (after 8 weeks). In the 9th week post-transduction, all cells failed to proliferate and metabolize, which led to the whole population’s death (Figure 5A). The doubling time in shRNA Control cells did not change over time and was constant and equal to 29 h.

The MDA-MB-231 cells initially showed longer (relative to MCF7) doubling time, i.e., 33 h. After the knockdown, this value increased to 36 h in the first week and then reached about 51 h (in week 9). The control vector showed no changes in the proliferation potential of studied cells (Figure 5A). Simultaneously, the Ki-67 protein level, a well-established proliferation marker, was also assessed. Incubation with 0.1 µM of doxorubicin was applied to reveal potential synergism of the hTERT silencing and the chemotherapeutic drug. Administration of DOX in shRNA Control MCF7 cells provoked a significant decrease of Ki-67 by about 50% (*p* < 0.05). The downregulation of hTERT provoked a very similar effect. Consequently, treatment of hTERT-downregulated cells with DOX provoked ca. a 70% decrease of Ki-67 level (*p* < 0.01) (Figure 5B,C). Similarly, the DOX administration provoked a 20% decrease of Ki-67 level in MDA-MB-231 shRNA Control cells, while downregulation of hTERT provoked about a 50% drop (*p* < 0.05). DOX applied to hTERT-downregulated cells caused about a 70% decrease of Ki-67 (*p* < 0.01) (Figure 5B,C).

Further, we assessed the tumor suppressor p53 protein level and its phosphorylated form (Ser15). The p53 protein is an essential mediator of many metabolic pathways in a cell, including DNA repair and apoptosis. It also contributes to the regulation of senescence and the cell cycle. The ATM-dependent phosphorylation of p53 (as a result of DNA damage and genomic destabilization) mediates activation of this protein at Ser15 that essentially leads to sequential series of additional phosphorylation events in p53 (including phosphorylation of Ser9-20, -46, and Thr18) that triggers further p53 induction and activation [40].

No changes in p53 protein level and phosphorylation at Ser15 position after hTERT downregulation were found in MCF7 cells (p53 wild type). On the other hand, a significant, approximately 2-fold increase (*p* < 0.001) in both assessed parameters was observed after the use of doxorubicin (0.1 µM), both in control (shRNA Control) and shRNA hTERT cells (MCF7 cell line; Figure 5B,C). Interestingly, in MDA-MB-231 cells (mutated p53), the level of p53, as well as p-p53, remained unaltered in all samples, i.e., in DOX-treated or in hTERT-downregulated cells. Similarly, no change in p53 accumulation and Ser15 phosphorylation was shown when shRNA hTERT cells were treated with DOX (0.1 μM) relative to shRNA Control (Figure 5B,C).

In the nucleus, p53 works as a transcriptional factor and regulates the transactivation of several proteins, including p21 [41]. In the nucleus, p21 binds to and inhibits the cyclin-dependent kinases, and blocks the transition from the G1 phase into S phase [42]. Numerous studies have shown that the p53/p21 pathway is involved in growth arrest, senescence, differentiation, apoptosis, or autophagy, depending on cell types and tissue contexts [43]. The analysis of the key cell cycle inhibitor, p21, revealed a significant increase in the accumulation of this protein in both cell lines treated with DOX (in MCF7 by 75%; *p* < 0.01, MDA-MB-231 by 35%; *p* < 0.05). Importantly, co-treatment provoked a cumulative effect in both cell lines, but the induction effect was more substantial in MDA-MB-231 cells (52%; *p* < 0.05 and 70%; *p* < 0.01) (Figure 5B,C). Noteworthy, the increased p21 level in MCF7 cells reflected the induced accumulation of those cells in the G0/G1 phase due to hTERT downregulation (Figure 4). Due to the observed increase of this protein, we performed the assessment of β-galactosidase (SA-β-Gal), i.e., 5-bromo-4-chloro-3-indole-β-D-galactopyranoside, another senescence marker. Interestingly, the enzyme’s slight cytoplasmic activity was observed only in the MCF7 shRNA hTERT cells, but no visible staining in the MDA MB-231 cells was observed (Figure 5D).

### 3.6. Identification of Autophagy in hTERT Downregulated MDA-MB-231 Cells

We observed some changes in the cell cycle of the MDA-MB-231 cells but no significant caspase-3 induction (Figure 4A,B) and moderate impact on proliferation (Figure 5A). From proliferation assessment, it appeared that hTERT downregulation triggered more effective cell survival inhibition in MCF7 cells. It was accompanied by the detection of a higher rate of G0/G1 cells accumulation, suggesting cell growth inhibition. Additionally, senescence markers, i.e., SA-β-Gal staining as well as p21 accumulation, were observed (Figure 5B–D), suggesting replicative senescence. In turn, in MDA-MB-231 cells, we observed a significantly lower antiproliferative effect of hTERT downregulation and no SA-β-Gal induction nor p21 alternation. Those observations implied the involvement of some other mechanisms, which suggested autophagy assessment. Thus, we decided to assess the impact of hTERT silencing on type 2 programmed cell death activation in MDA-MB-231 cells. For this reason, we verified levels of the proteins crucial for the regulation and execution of autophagy. It is a mechanism that is activated under stressful conditions and can be related to the regulation of the mTORC1 complex composed of mTOR kinase, Raptor protein, and mLST8/GβL protein. DOX treatment did not affect the main macroautophagy regulator, mTOR level. The assessment of this protein in shRNA hTERT-treated MDA-MB-231 cells revealed a slight but significant decrease by about 30% (*p* < 0.05) (Figure 6A,B). A similar observation was made in the cells treated with both DOX and shRNA hTERT. The analysis of beclin-1, which is perceived as one of the critical autophagy markers (involved in the initial stage), revealed a 40% increase (*p* < 0.05) in hTERT-downregulated cells as well as in cells co-treated with DOX. No such effect was observed in cells treated with DOX alone. Assessment of Atg5 (involved in phagophore elongation) showed that the DOX treatment provoked accumulation of this protein, and hTERT downregulation increased this effect up to over 40% (*p* < 0.01). Similarly, a smaller but also significant increase of this protein was observed in cells treated with a combined DOX and shRNA hTERT. At the same time, DOX alone did not show any significant alteration. The evidence of the formation of autophagosomes in the cell, the LC3 II /I ratio, was increased by approx. 70% (*p* < 0.01) after hTERT downregulation, while a combined treatment with DOX resulted in only a 50% increase (*p* < 0.05). Noteworthy, DOX treatment did not provoke alteration in the ratio. The final stage of autophagy is the fusion of the autophagosomes with the lysosome, which results in the degradation of the p62 protein. This effect was observed after hTERT downregulation alone or after the combination with DOX (> 50% decrease, *p* < 0.01 and 40%, *p* < 0.05, respectively). DOX alone did not provoke any significant change in p62 accumulation (Figure 6A,B).

### 3.7. Contribution of hTERT Downregulation to Cancer Cells Motility

To identify the molecular pathways involved in increasing the sensitivity of cancer cells to chemotherapeutic agents after hTERT silencing, we performed an adhesion test and scratch assay. The tests have been chosen based on the observed metabolic changes after hTERT downregulation in both breast cancer cell lines. Matrigel^®^ was used in the adhesion assay to provide optimal conditions for attachment, mimicking the presence of an extracellular matrix. Importantly, impaired attachment in tested cells was observed. When the medium and cell suspension were removed 3 h after seeding (time interval sufficient for the attachment of control cells), a decrease in the number of hTERT shRNA cells anchored at the bottom of the plate was reported. These observations indicated a disturbance of the adhesion process in a large part of this population in both tested cell lines, relative to control cells, i.e., 20% in MCF and 30% decrease of attached MDA-MB-231 cells (Figure 7A). 

In turn, to verify the migration modulation, we performed a wound-healing assay. This assay also indicated that hTERT downregulation, leading to serious cancer cell metabolism dysfunctions, attenuated in the migratory potential of the cells after 24 h, but with no significance (Figure 7B). This effect was more substantial in MDA-MB-231 cells than in MCF7. Longer incubation time intervals revealed a significant difference in the number of cells migrating into the wound in both cell lines (i.e., 48 and 72 h). However, it led to a conclusion that this might result not only from migration impairment but also from the alteration in proliferation since the doubling time of both cell lines was exceeded. Noteworthy, it corresponds to the results observed in the proliferation assay (Figure 5A). 

Significantly, the most crucial pathways in cell communication, adhesion, and migration are associated with the signaling of β1-integrin. Interestingly, we observed a significant decrease of the key players in this pathway, i.e., β1-integrin, FAK, paxillin, and c-Src, as well as disruption in their phosphorylation in MCF7 shRNA hTERT cells (Figure 7C). After hTERT downregulation, we observed a decrease in the accumulation of β1-integrin and FAK in both cell lines. 

Additionally, a significant decrease of FAK phosphorylation was noted at both sites, i.e., Tyr 397 and Tyr 576/577. However, co-treatment with DOX did not cause any additive effect. In turn, when cells were treated with DOX alone, FAK was downregulated but only in MCF7 with no alteration in MDA-MB-231 cells. Interestingly, FAK phosphorylation in MCF7 at Tyr 397 was unaltered, while at Tyr 576/577 was significantly induced. In MDA-MB-231 cells, after DOX treatment, the phosphorylation status of FAK at Tyr 397 and Tyr 576/577 was significantly induced. Assessment of paxillin revealed that the basal protein level was decreased after hTERT downregulation, and additional DOX treatment did not affect this response. 

In contrast, the treatment of cells with the drug alone did not provoke any significant paxillin alterations relative to control. This effect was similar in both cell lines. The phosphorylated paxillin demonstrated significantly reduced levels after hTERT downregulation, and an additive effect was observed after combination with DOX. 

Immunoidentification of another FAK-associated protein, c-Src, showed significant downregulation of this protein in MCF7 after hTERT downregulation, and when a combination with DOX was applied. The drug alone did not provoke any significant modulation. In MDA-MB-231, no significant alteration in c-Src was observed. Interestingly, Src phosphorylation assessment showed a significant increase of p-Src (Tyr 527) in both cell lines after hTERT downregulation and after drug co-treatment. DOX alone did not cause any significant difference. In turn, assessment of the p-Src (Tyr 416) showed a significant decrease in both cell lines after hTERT downregulation or in combination with DOX in both cell lines. Treatment with DOX alone did not show any significant changes relative to control cells in none of the cell lines.

## 4. Discussion

Telomerase plays a crucial role in acquiring cancer phenotype by aiding the unlimited replicative potential of cancer cells. It was reported that hTERT had a telomere-independent role in cancer progression through an unknown mechanism [44]. To investigate the molecular function of the key telomerase subunit in cancer cells, we performed the silencing of hTERT in breast cancer cells using shRNAs. Obtained data demonstrated a significant reduction in hTERT gene expression at the transcriptional and protein levels in both cell lines, together with a decrease in telomerase activity. Additionally, we analyzed the effect of hTERT downregulation on the response of human breast adenocarcinoma cell lines MCF7 and MDA-MB-231 to combined treatment with a chemotherapeutic drug, DOX. The idea of our study was to explore whether long-term RNA interference directed against hTERT could alter breast cancer cells’ survival, drug sensitivity, proliferation, and adhesion abilities.

### 4.1. hTERT Downregulation and Breast Cancer Cells Survival

Our study revealed that the reduction of hTERT expression provoked a significant diminishment in breast cancer cells’ survival. Furthermore, MTT and clonogenic results also showed that hTERT downregulation led to cancer cell sensitization to doxorubicin. Similar findings have been obtained by Liu et al. (2013) in the same experimental model, using RNAi and adenoviruses, and by Cerone et al. (2006) in two breast cancer cell lines with different p53 and estrogen receptor status [45,46]. Studies performed in other cancer cell types show similar results [47,48,49]. Interestingly, telomerase inhibition was shown to sensitize breast cancer cells to various drugs with different mechanisms of action [50], which implies a broad spectrum of pathways being controlled by (or associated with) hTERT. Fleisig et al. (2016) propound that expression of this subunit facilitates cell growth and proliferation after chemical- or metabolism-induced genotoxic stress [11]. This protective function of hTERT is uncoupled from its role in telomere synthesis and structure maintenance. Data show that hTERT expression provides oncogenic transformed cells with survival advantages by sheltering them from double-strand DNA breaks [11]. It was also reported that hTERT controlled the expression of various genes (e.g., NF-κB-dependent gene expression) implicated in the control of cell proliferation and cancer progression, proposing that it might act as an oncogene in a telomere-independent manner [24]. Additionally, hTERT silencing was shown to sensitize cancer cells to chemotherapeutic drugs; consequently, genetic, pharmacological, and antisense methods of telomerase inhibition have been developed [51,52,53,54]. The use of telomerase-specific inhibitors, e.g., GRN163L or BIBR 1532, provides some inconsistent results due to different mechanisms of action base on telomerase-telomere interactions [50,55]. That suggested some non-canonical functions of hTERT that would not depend on telomere binding or attrition but would still affect cell survival. However, it is unclear if the protective functions of hTERT require its catalytic activity [56,57]. Our data indicated sensitization to DOX (reactive species inducer) due to hTERT downregulation. Notably, one of the critical non-canonical functions of hTERT in cell survival and aging is mitochondrial protection under increased oxidative stress. Under these conditions, the hTERT subunit is excluded from the nucleus and translocated into the mitochondria, which appears to serve as an anti-apoptotic mechanism [10,12,57,58]. The results of Nakamura et al. concerning the assessment of DNA double-strand breaks inducing mechanisms suggest a specific interaction between hTERT and the DNA repair process in human cells [48]. However, the ultimate effect may depend on many individual factors, including heterogeneity of cancer and a patient’s genetic profile, making the therapy efficacy challenging to predict. Thus, the further study is still required to advance our understanding of this phenomenon.

In our long-term study, we observed a significant diminishment of the population doubling level after telomerase downregulation. The MCF7 cells died 9 weeks after transduction, and MDA-MB-231 cells showed prolonged doubling time but continued to divide until the final examination, i.e., 64 days. The suppression of hTERT expression inhibited the proliferation of breast cancer cells, but these cells did not exhibit immediate cell death. According to literature data, reduced hTERT expression may result in cell cycle arrest either in phase G1 or S/G2, depending on the cell line [51]. Our data showed an increase in G0/G1 and decreased S/G2 cell population in MCF7 cells. Similar results were obtained in the pancreatic cancer model, liver, and studies on cervical cancer [29,59,60]. In our study, in MDA-MB-231 cells, the fraction of the dead cells was slightly elevated. Additionally, in both cell lines, Ki-67, a well-known proliferation marker of the G1 stage [61], was significantly decreased. Some investigators reported that the contribution of hTERT to cell survival was cell cycle stage-specific [55]. Our result showed no activation of the p53 pathway (neither basal p53 nor Ser-15 phosphorylated form was altered), but a significant accumulation of p21 was observed in both cell lines after hTERT downregulation. It suggested a possible association with PI3K that may directly affect p21 [62].

Importantly, MDA-MB-231 cells are reported to carry a mutation in the exon 8, codon 280 of the tumor suppressor gene [39], which probably explains the minor p53 protein response in these cells. The mutation leads to the loss of Bcl-XL and Bcl-2 binding domain, thus losing cytochrome c release from mitochondria, leading to at least partial loss of cytoplasmic pro-apoptotic activity [63,64]. This phenotype may disturb apoptosis or senescence activation [65]. Possibly, it may be associated with a pro-survival action of autophagy in cancer cells. Interestingly, we revealed autophagy activation in MDA-MB-231 cells after hTERT downregulation, manifested by Atg 5, Beclin 1, LC3 II/ LC3 I induced accumulation, and p62 significant reduction. Altogether, it may suggest another protective mechanism that promotes cancer cell survival (as suggested by Lee at al., [56]) and may explain significantly slower proliferation inhibition after hTERT downregulation in these cells.

Additionally, MDA-MB-231 cells were reported to show longer telomeres than MCF7 cells [66]. It could explain the different responses of both cell lines and higher sensitivity of MCF7 cells to hTERT downregulation (proliferation assessment, SA-β-galactosidase staining, cell cycle arrest at the G0/G1 phase, and no apoptosis) than MDA-MB-231 cells. Moreover, we demonstrated that telomere attrition in MCF7 was higher after hTERT downregulation than in MDA-MB cells, which might indicate a telomere crisis in these cells, as previously suggested [67].

### 4.2. hTERT Downregulation and Breast Cancer Cells Adhesion and Migration

Until now, only a few studies have reported the promotion of cell invasion as novel non-canonical functions of hTERT. This type of engagement appears to act through the upregulation of metalloproteinases (MMPs) independently of telomerase activity [68]. Consequently, it triggers pathways closely related to tumorigenesis, metastasis, and invasion. Our study revealed that hTERT downregulation correlated with a significant decrease in breast cancer cell migration and adhesion. It was demonstrated using the scratch assay, which suggested a significant proliferation decrease when performed in a long-term assessment (48 and 72 h). Migration alteration was manifested by remarkable variation in the level of individual proteins (β1-integrin, paxillin, c-Src, FAK) together with the reduction of their phosphorylation and reduced survival and metastatic potential. Two latest publications support this observation from Liu et al. and Magissano et al. that suggest such association [26,30]. The first paper revealed an association between telomerase/hTERT expression and human bone osteosarcoma U2OS cells’ migration and adhesion ability. This observation was important since U2OS cells are deprived of telomerase activity. hTERT was delivered via transfection, and it provoked adhesion and migration promotion in those cells. Described phenomena were accompanied by alterations in the expression of PDPN, SPP1, BARX2, and MMPs, which are associated with the remodeling of the extracellular matrix [26]. The second study demonstrated that the silencing of hTERT reduced cell proliferation and caused a decrease in invasion and migration ability of three human anaplastic thyroid cancer cell lines CAL-62, 8505C, and SW1736 without affecting the telomere length. Importantly, they used siRNA for hTERT downregulation, which implicates no contribution of telomere attrition to the observed effects due to short-time experiments [30].

## 5. Conclusions

Demonstrated results provide new evidence to support the broad spectrum of hTERT functions. Our data contribute to understanding telomerase metabolism and increase cancer cell elimination efficiency, especially the most difficult ones to eliminate, i.e., resistant to drugs and more aggressive. However, since we observed induction of autophagy in MDA-MB-231 cells, it must be considered that in some cases, hTERT elimination may lead to autophagy. This approach may provoke cell death or survival. Consequently, a proper adjuvant therapy must be designed to push cancer cells to the death pathway. However, as suggested, telomerase activity or telomere length maintenance does not seem to be a critical issue for cancer cells’ resistance. Nevertheless, it may be that hTERT on its own, or together with other proteins or adhesion and migration molecules, play a crucial role in this process. Our findings confirmed that cancer cells deprived of hTERT are more vulnerable and susceptible to cancer drugs that may eventually lead to autophagy. The association between autophagy and telomerase/hTERT requires further detailed studies. It may be that the ultimate effect could be cell-type specific or dependent on stress conditions, metabolic impairment, but also the basal level of hTERT. Such correlation requires broad investigation using modulators of hTERT and autophagy, including several intermediate pathways. One should not forget about the ability of cancer cells to induce the Alternative Telomere Lengthening (ALT) that may be induced in some cancer cells after hTERT downregulation or drug treatment [69]. Assuming the protective role of hTERT in cancer cell response to stress it may be that via contribution to cell signaling pathways this factor affects the autophagy induction. Consequently, depending on the basal or induced hTERT level or telomere length, the ultimate metabolic effect could be different as already suggested [70,71], i.e., pro- or anti-survival. Elucidating the mechanism of non-canonical functions of hTERT is an important step in developing diagnostic and therapeutic applications. It provides a new direction for the search for novel telomerase key subunit-associated pathways.

## Figures and Tables

**Figure 1 cells-10-00867-f001:**
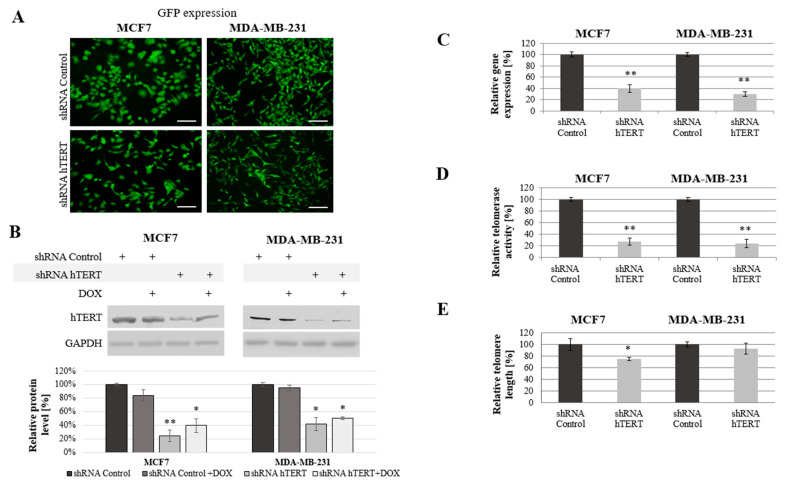
Analysis of human telomerase reverse transcriptase (hTERT) downregulation effect in MCF7 and MDA-MB-231 cells. (**A**) Representative images of GFP expression in the MCF7 and MDA-MB-231 cells transduced with pLV-THEM-shTERT (shRNA hTERT) and pLV-THEM-shRNA (shRNA Control); magnification 40×, scale bars 100 μm; (**B**) hTERT gene expression at the protein level, semi-quantitative analysis of Western blot results using Image Studio Lite software. GAPDH was used as loading control. (**C**) Relative hTERT gene expression at the transcriptional level measured by qPCR. (**D**) Relative telomerase activity determined by telomeric repeat amplification protocol (TRAP) assay. (**E**) Relative telomere length measured by qPCR. shRNA Control, cells transduced with lentiviral vectors containing scramble shRNA; shRNA hTERT, cells transduced with lentiviral vectors containing shRNA against hTERT. 0.1 μM doxorubicin (DOX); 8h treatment. Data are presented as the mean ± standard deviation. (*) the symbol for *p* < 0.05; (**) for *p* < 0.01, with comparison to shRNA Control.

**Figure 2 cells-10-00867-f002:**
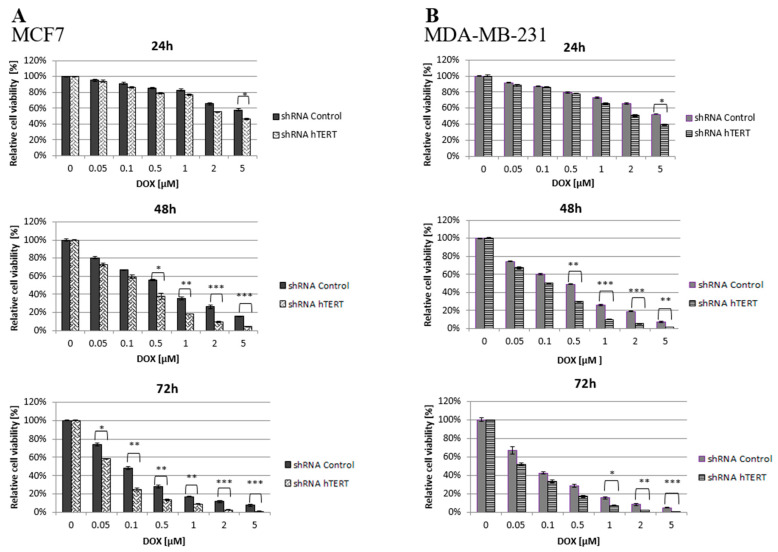
Cell survival of hTERT-downregulated MCF7 (**A**) and MDA-MB-231 (**B**) cells after treatment with doxorubicin determined by MTT. Breast cancer cells were subjected to treatment with DOX on the 21st-day post-transduction. Data are presented as the mean ± standard deviation. (*) *p* < 0.05; (**) *p* < 0.01; (***) *p* < 0.001 relative to shRNA Control cells.

**Figure 3 cells-10-00867-f003:**
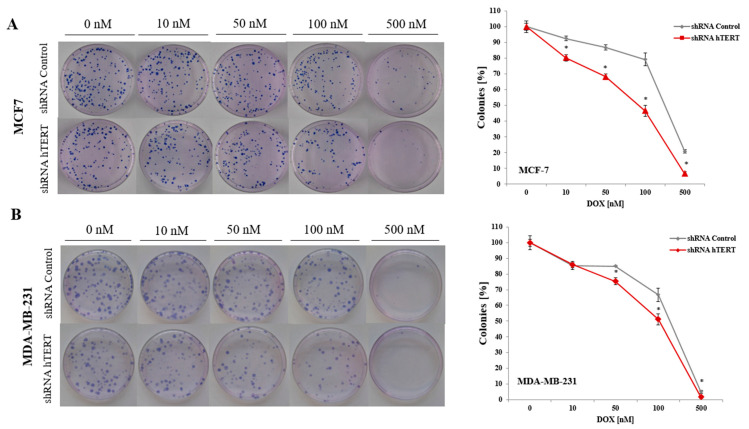
Colony formation capacity of breast cancer cells after hTERT downregulation in a long-term study. The clonogenic potential of MCF7 (**A**) and MDA-MB-231 (**B**) after treatment with doxorubicin was determined. Cells were subjected to experiment on 21^st^-day post-transduction. The representative images of single-cell clone proliferation, stained with Giemsa’s solution. Data are presented as the mean ± standard deviation. (*) *p* < 0.05 relative to shRNA Control.

**Figure 4 cells-10-00867-f004:**
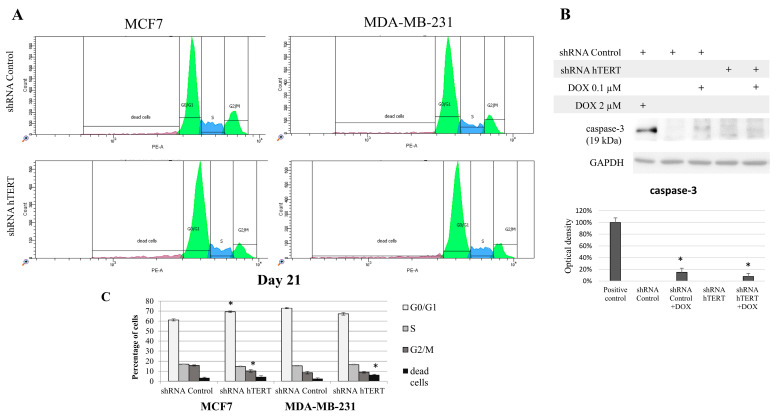
The contribution of hTERT downregulation to cell cycle modification of breast cancer cells. (**A**) Propidium iodide labeling and representative flow cytometry histograms of cell cycle analysis. Both studied cell lines, i.e., MCF7 and MDA-MB-231 were subjected to the cell cycle analysis 21 days after transduction. (**B**) Graphical representation of the histogram results. Data are expressed as the mean ± SD of the data obtained from at least three independent experiments. * *p* < 0.05. (**C**) Analysis of caspase-3 involved in I programmed cell death in MDA-MB-231 cells. Immunodetection of cleaved caspase-3 was performed using Western blot on 21st-day post-transduction, after 8 h DOX (0.1 μM) treatment; densitometry analysis was performed out of three scanned membranes from three independent experiments; 2 μM DOX was used as a positive apoptosis control (8 h treatment). (*) *p* < 0.05 relative to shRNA Control.

**Figure 5 cells-10-00867-f005:**
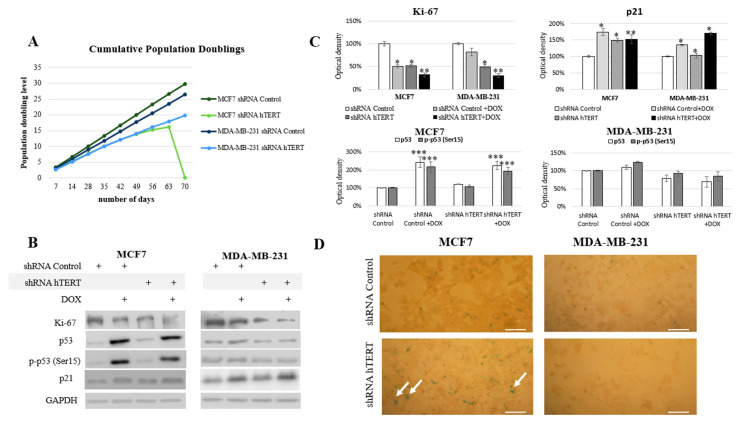
Telomerase downregulation decreases the proliferation of breast cancer cells. (**A**) The cumulative population doublings was calculated for each 96 h ((log (total number of cells counted at the day of passage) − log (number of cells initially seeded at the previous passage))/log2). The monitoring period of proliferation potential is 9 weeks after transduction. The experiment was repeated three times. (**B**) Immunodetection of Ki-67, p53, p-p53, and p21 was performed using Western blot in MCF7 and MDA-MB-231 cells on the 21st-day. 0.1 μM DOX; 8h treatment, followed by densitometry analysis (**C**). (*) *p* < 0.05; (**) *p* < 0.01; (***) *p* < 0.001 relative to shRNA Control. Densitometry analysis was performed out of three scanned membranes from 3 independent experiments. (**D**) Analysis of the biochemical-aging marker, the enzyme β-galactosidase (SA-β-Gal). Cells were subjected to hTERT downregulation, and the effect was assessed after 21 days from transduction. A typical result out of 2 replicates was demonstrated (arrows indicate green/β-galactosidase signal), magnification 100×, scale bars 100 μm.

**Figure 6 cells-10-00867-f006:**
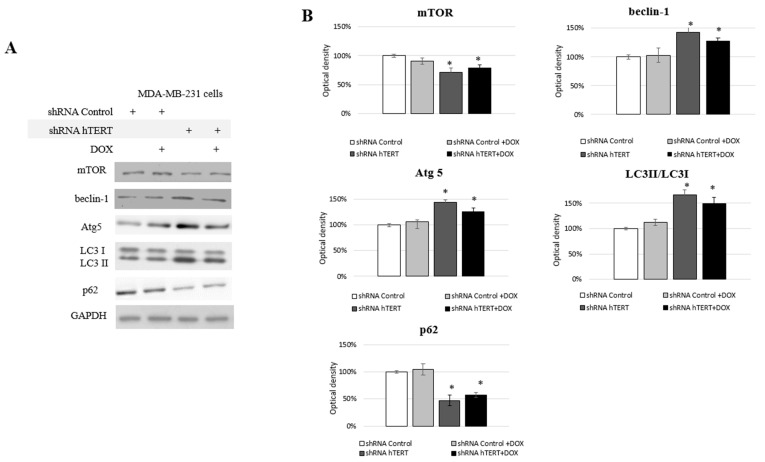
Analysis of proteins involved in II programmed cell death in MDA-MB-231 cells. Immunodetection of mTOR, beclin-1, Atg5, LC3I/LC3II, p62 was performed using Western blot on 21st-day post-transduction, 0.1 μM DOX; 8 h treatment (**A**); followed by densitometry analysis (**B**). (*) *p* < 0.05 relative to shRNA Control. Densitometry analysis was performed out of three scanned membranes from three independent experiments.

**Figure 7 cells-10-00867-f007:**
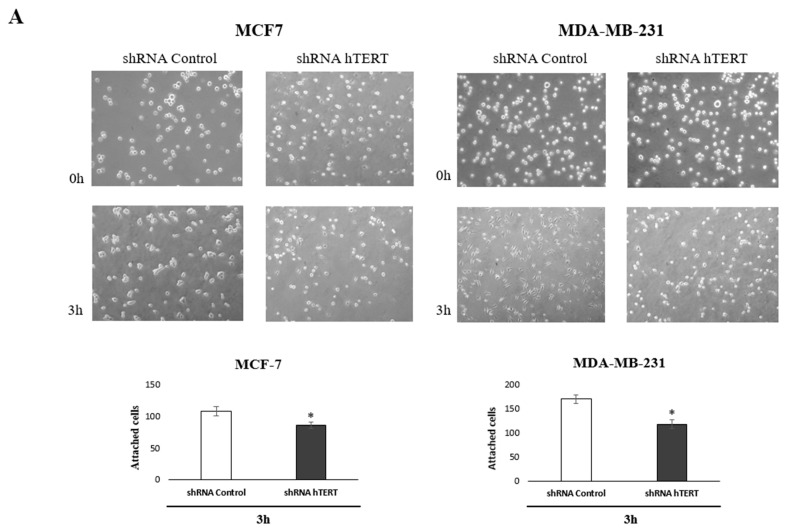
Contribution of hTERT downregulation to functional impairment of migration and adhesion in MCF7 and MDA-MB-231 cells. (**A**) Adhesion assay. The photos are representative of three independent experiments (magnification 40×). Adhesion cells were counted in at least three fields in each well. Each bar represents the mean ± SD of the data obtained from three independent experiments. (*) *p* < 0.05 relative to shRNA Control. (**B**) Wound-healing migration assay. Cells were scraped with the pipette tip. The photos represent cell migration under the microscope at 100× magnification field after the injury. A typical result out of three replicates was demonstrated. The migration of studied cells was quantified by measuring wound closure areas after injury. Each bar represents mean ± SD (n = 3). (*) *p* < 0.05 relative to shRNA Control cells (**C**) immunodetection of β1-integrin, FAK, p-FAK (Tyr397), p-FAK (Tyr576/577), paxillin, p-paxillin, Src, p-Src (Tyr527), and p-Src (Tyr416) was performed 21st-day post-transduction, using Western blot; 0.1 μM DOX; 8 h treatment, followed by densitometry analysis. (*) *p* < 0.05; (**) *p* < 0.01 relative to shRNA Control. Densitometry analysis was performed on three scanned membranes from three independent experiments.

## Data Availability

The data presented in this study are available on request from the corresponding author. The data are not publicly available due to the large number and size of the files.

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
