# Peer review of "hTERT Downregulation Attenuates Resistance to DOX, Impairs FAK-Mediated Adhesion, and Leads to Autophagy Induction in Breast Cancer Cells"

_cells, 2021, doi:10.3390/cells10040867_

Round 1
Reviewer 1 Report
In this study, the authors investigated the effect of hTERT downregulation alone or combined with DOX treatment in two breast cancer cell lines. They showed that hTERT downregulation had an effect on several pathways such as cell migration and adhesion and autophagy.
Specific comments:
1/ The long-term effect of hTERT downregulation in both cell lines cannot be uncoupled from telomere erosion. Telomere length should be measured 21 days post-transduction and compared to the initial telomere length.
For long term monitoring of the proliferation, telomere length should also be measured. The cell death observed after 9 weeks is probably due to a telomere like-crisis (see baird DM et al, Cell report 2014).
3/ Figure 5a: Show the growth curves displaying the cumulative PDs as a function of the cumulative number of days. To calculate the cumulative PDs use the following formula: [log (total number of cells counted at day of passage) – log (number of cells initially seeded at previous passage)]/log2. Cumulative Population Doubling is the total number of PDs at a given day point.
2/ The beta galactosidase experiment should be shown.
3/ Show representative images of the plates for the clonogenicity assay.
4/ Title 3.5: the authors should not talk about senescence since senescence was not assessed. Classically, an inhibition of telomerase in cancer cells leads to critical telomere shortening and a telomere-crisis like state.
General comments and minor issues:
1/ In general, check the language and use more precise vocabulary. For example: line 43-44 instead of “restoration of telomeres «write “maintenance of telomere length”. Line 45 instead of “this structure degrades” write “this structures shortens” and so on.
2/ Line 41: add references
3/ Line 42-44: idem add references
4/ Line 47-48: the statement that replicative senescence leads to cell death is incorrect. Replicative senescence is a stable cell cycle arrest. Senescence and cell death are distinct pathways.
5/ Line 76: precise what you mean when referring to “telomeric subunits”.
6/ Line 332 what do the authors mean by cell reproductive death ?
Author Response
Dear Reviewer,
first, we would like to thank the reviewers for the thorough revision and constructive comments. Please find below our response. To emphasize, the presented work is a part of a greater project that is in progress, and we hope to resolve the mechanistic aspects in the field of telomerase-autophagy association in the nearest future. Unfortunately, we could not address all the suggested issues within the imposed term of 10 days since most of our experiments require 21 days of treatment and further processing. All the suggested amendments were enrolled and marked with red color.

Reviewer 2 Report
In the paper entitled “hTERT downregulation attenuates resistance to Dox, impairs 2 FAK-mediated adhesion, and leads to autophagy induction in breast cancer cells”, the authors aimed to explore how telomerase downregulation in breast cancer cells affect their ability to survive. They demonstrate that long-term TERT downregulation does not affect breast cancer cell survival but rather render them more sensitive to Doxorubicine. They also showed that TERT downregulation decreases their proliferation and migration, and increases their capacity of activating autophagy.
The general take-home message is not new, but the study is well done and supports the telomere-independent function of TERT on proliferation, migration of cancer cells, resistance to chemotherapy and autophagy that has been described elsewhere, as cited by the authors.
The most interesting parts lie in the two last figures (7 and 6) which demonstrate that TERT downregulation impairs FAK-mediated adhesion and induces autophagy. However, the authors could have explored the autophagy topic a little bit more because the biological consequence of their finding is unclear.
Figure 6:
- Why were the autophagy experiments not performed in MCF7 tumor cell line as well (like for the other figures)? Autophagy can influence the sensitivity to chemotherapy (I don't think the authors clearly mentionned this point), and the data indicate that shRNA TERT MCF7 cells are the most sensitive to Doxorubicine.
- The exact mechanism by which autophagy drives survival, cell death or resistance to treatment is still a tricky issue. In their discussion, the authors assumed that autophagy induced by TERT downregulation in MDA cells may promote cancer cell survival rather than cell death. What about the relationship between high TERT level, autophagy and cell survival? Could it be that the increase of autophagy in cancer cells with low TERT expression is rather responsible for their senescence (Slobodnyuk K, Cell death and disease, 2019) or their sensitivity to chemotherapy as discussed in the review by Li X et al (Biomedicine & Pharmacotherapy, 2019)? Using autophagy inhibitors (e.g chloroquine) to limit the autophagy flux in shRNA TERT MCF7 and MDA6-MB-231 cancer cells and address the consequence on survival/resistance to Doxorubicine could possibly help clarify this point.
Author Response

(The authors gave the same response as above.)

Round 2
Reviewer 1 Report
I do think that telomere length should be measured as it is already known that downregulation of telomerase in cancer cells leads to a telomere crisis marked by massive cell death (Jones R et al. cell report 2014). Moreover, the differences observed by the authors between the two cell lines might be due to a difference in the initial telomere length. If the telomeres in the MDA cell line are initially longer that would explain the differences observed at 21 days post-transduction between the cell lines in term of proliferation. The authors indicated that they did try to assess telomere length by Q-PCR but the that the results were not conclusive. They could instead use the TRF method. If they still have some DNA samples, they would not need to do again the transduction and wait for 21 days.
Author Response
Dear Reviewer,
one more time, we would like to thank you for the thorough revision and constructive comments. Please find below our response.
All the amendments enrolled are marked with red color.
Specific comments:
I do think that telomere length should be measured as it is already known that downregulation of telomerase in cancer cells leads to a telomere crisis marked by massive cell death (Jones R et al. cell report 2014). Moreover, the differences observed by the authors between the two cell lines might be due to a difference in the initial telomere length. If the telomeres in the MDA cell line are initially longer that would explain the differences observed at 21 days post-transduction between the cell lines in term of proliferation. The authors indicated that they did try to assess telomere length by Q-PCR but the that the results were not conclusive. They could instead use the TRF method. If they still have some DNA samples, they would not need to do again the transduction and wait for 21 days.
RESPONSE:
We started the experiments during the first round of revision that enabled another measurement of telomere length with bigger amount of DNA and optimization of the reaction conditions. Telomere length was measured with the method currently available in our lab i.e. qPCR that is based on Cawthon (Cawthon, Nucleic Acids Res. 2009).
Consequently, the Figure 1, the Materials and Methods section, the Results section as well as the Discussion section were supplemented with respective data as follows:
Figure 1 was amended and attached in the present form as below.
Figure 1. Analysis of hTERT downregulation effect in MCF7 and MDA-MB-231 cells. (A) Representative images of GFP expression in the MCF7 and MDA-MB-231 cells transduced with pLV-THEM-shTERT (shRNA hTERT) and pLV-THEM-shRNA (shRNA Control); magnification 40 x, scale bars 100 μm; (B) hTERT gene expression at the protein level, semi-quantitative analysis of Western blot results using Image Studio Lite software. GAPDH was used as loading control. (C) Relative hTERT gene expression at the transcriptional level measured by qPCR. (D) Relative telomerase activity determined by TRAP assay. (E) Relative telomere length measured by qPCR. shRNA Control, cells transduced with lentiviral vectors containing scramble shRNA; shRNA hTERT, cells transduced with lentiviral vectors containing shRNA against hTERT. 0.1 μM DOX (doxorubicin); 8h treatment. Data are presented as the mean ± standard deviation. (*) the symbol for p<0.05; (**) for p<0.01, with comparison to shRNA Control.
Materials and Methods
2.3.1. Relative telomere length assessment using qPCR
DNA was extracted from cacer cells (1x106 cells in each sample) after hTERT downregulation using a Genomic Mini DNA Isolation kit (A&A Biotechnology, Poland). A high concentration sample of genomic DNA was prepared in decimal concentrations that were used to run as a standard curve. Telomere length was assessed using two pairs of primers, specific towards telomeres (Telg: ACACTAAGGTTTGGGTTTGGGTTTGGGTTTGGGTTAGTGT and Telc: TGTTAGGTATCCCTATCCCTATCCCTATCCCTATCCCTAACA) [34] and single copy gene, albumin (ALBF: TTTGCAGATGTCAGTGAAAGAGA and ALBR: TGGGGAGGCTATAGAAAATAAGG), as previously described [35]. Briefly, the conditions were optimized as follows: 95°C for 10 min, followed by two cycles of 94°C for 15 sec and 49°C for 15 sec without fluorescence acquisition and 40 cycles (94°C for 10 sec, 61°C for 10 sec and 72°C for 10 sec) with signal acquisition. For albumin gene copies the conditions were optimized as follows: denaturation at 95°C for 10 min, followed by 45 cycles at 94°C for 10 sec, 61°C for 10 sec and 72°C for 10 sec. The MgCl2 was 2.5 mM in both reactions while the primer concentration was 0.9 or 0.5 µM for telomere or albumin copies assessment, respectively. Melting analysis (range, 65–95°C; resolution, 0.2°C) was performed to verify the specificity of the products. The efficiency of the reactions was higher than 95%. The assay was performed using the LightCycler® 2.0 Instrument and the LightCycler® FastStart DNA Master SYBR Green I kit (Roche Diagnostics).
- Cawthon RM. Telomere length measurement by a novel monochrome multiplex quantitative PCR method. Nucleic Acids Res. 2009;37(3):e21. doi:10.1093/nar/gkn1027
- Barczak W, Rozwadowska N, Romaniuk A, Lipińska N, Lisiak N, Grodecka-Gazdecka S, Książek K, Rubiś B. Telomere length assessment in leukocytes presents potential diagnostic value in patients with breast cancer. Oncol Lett. 2016;11(3):2305–2309. doi:10.3892/ol.2016.4188
Results
(…) hTERT downregulation and cell growth were being monitored for up to 45 days. However, all the experiments were performed on day 21 from transduction, as mentioned in the Materials and Methods section. Since some reports postulate the additive effect of therapeutic agents and telomerase modulators in cancer cells, we included tests using a combination of hTERT downregulation and doxorubicin. The concentration of the drug was carefully selected based on previous MTT experiments [36], and it was 0.1 μM.
A significant reduction of hTERT protein level at 75% (p<0.01) in MCF7 and 60% (p<0.05) in MDA-MB-231 cells was observed after hTERT gene downregulation (Figure 1B). Importantly, doxorubicin alone in control cells or shRNA hTERT cells did not cause any significant effect in hTERT expression (western blot) when subjected to the treatment in both tested cell lines (Figure 1B). Similarly, no additive effect in the level of protein accumulation was observed. Further, hTERT downregulation was shown effective in reducing the hTERT transcript level in both MCF7 and MDA-MB-231 cells (reduction by 60%, p<0.01 and 70%, p<0.01, relative to mock shRNA, respectively) (Figure 1C). Similarly, telomerase assessment by TRAP assay revealed a significant decrease of the enzyme activity in both cell lines >75% after hTERT downregulation (p<0.01, Figure 1D).
We expected that telomerase downregulation would eventually lead to telomeres attrition so we performed the assessment of their length dynamics. After 21 days from transduction, telomere length in MCF7 cells was significantly reduced by more than 25% relative to shRNA Control sample. In MDA-MB-231 cells, the reduction was not significant (lower than 10%) but also noticeable (Figure 1E).
Discussion
Additionally, MDA-MB-231 cells were reported to show longer telomeres than MCF7 cells [66]. It could explain the different responses of both cell lines and higher sensitivity of MCF7 cells to hTERT downregulation (proliferation assessment, SA-β-galactosidase staining, cell cycle arrest at the G0/G1 phase, and no apoptosis) than MDA-MB-231 cells. Moreover, we demonstrated that telomere attrition in MCF7 was higher after hTERT downregulation than in MDA-MB cells which might indicate a telomere crisis in these cells, as previously suggested [67].
- Vera E, Canela A, Fraga MF, Esteller M, Blasco MA. Epigenetic regulation of
telomeres in human cancer. Oncogene. 2008;27(54):6817-33. doi:10.1038/onc.2008.289.
- Jones RE, Oh S, Grimstead JW, Zimbric J, Roger L, Heppel NH, Ashelford KE, Liddiard K, Hendrickson EA, Baird DM. Escape from telomere-driven crisis is DNA ligase III dependent. Cell Rep. 2014;8(4):1063-76. doi:10.1016/j.celrep.2014.07.007.